# The Influence of a Knitted Hydrophilic Prosthesis of Blood Vessels on the Activation of Coagulation System—In Vitro Study

**DOI:** 10.3390/nano11061600

**Published:** 2021-06-18

**Authors:** Maria Szymonowicz, Maciej Dobrzynski, Sara Targonska, Agnieszka Rusak, Zbigniew Rybak, Marcin H. Struszczyk, Jacek Majda, Damian Szymanski, Rafal J. Wiglusz

**Affiliations:** 1Department of Experimental Surgery and Biomaterials Research, Wrocalw Medical University, Bujwida 44, 50-368 Wroclaw, Poland; maria.szymonowicz@umed.wroc.pl (M.S.); zbigniew.rybak@umed.wroc.pl (Z.R.); 2Department of Pedriatric Dentistry and Preclinical Dentistry, Wroclaw Medical University, Krakowska 26, 50-425 Wroclaw, Poland; maciej.dobrzynski@umed.wroc.pl; 3Institute of Low Temperature and Structure Research, Polish Academy of Sciences, Okolna 2, 50-422 Wroclaw, Poland; s.targonska@intibs.pl (S.T.); d.szymanski@intibs.pl (D.S.); 4Department of Histology and Embryology, Wroclaw Medical University, T. Chalubinskiego 6a, 50-368 Wroclaw, Poland; agnieszka.rusak@umed.wroc.pl; 5Institute of Security Technologies MORATEX, Marii Sklodowskiej-Curie 3, 90-505 Lodz, Poland; mstruszczyk@moratex.eu; 6Department of Laboratory Diagnostics, 4th Military Clinical Hospital, R. Weigla 5, 50-981 Wroclaw, Poland; jacek_majda@interia.pl

**Keywords:** hydrophobic and hydrophilic vascular prostheses, spectroscopy study, differential scanning calorimetry, scanning electron microscopy, factors activity coagulation, coagulation system activation

## Abstract

The replacement of affected blood vessels of the polymer material can cause imbalances in the blood haemostatic system. Changes in blood after the implantation of vascular grafts depend not only on the chemical composition but also on the degree of surface wettability. The Dallon^®^ H unsealed hydrophilic knitted vascular prosthesis double velour was assessed at work and compare with hydrophobic vascular prosthesis Dallon^®^. Spectrophotometric studies were performed in the infrared and differential scanning calorimetry, which confirmed the effectiveness of the process of modifying vascular prostheses. Determination of the parameters of coagulation time of blood after contact in vitro with Dallon^®^ H vascular prosthesis was also carried out. Prolongation of activated thromboplastin time, decreased activity of factor XII, IX and VIII, were observed. The prolonged thrombin and fibrinogen were reduced in the initial period of the experiment. The activity of plasminogen and antithrombin III and protein C were at the level of control value. The observed changes in the values of determined parameters blood coagulation do not exceed the range of referential values for those indexes. The observed changes are the result of considerable blood absorptiveness by the prosthesis of blood vessels and their sealing.

## 1. Introduction

The possibility of the use of natural or synthetic vascular implants is one of the achievements in angiosurgery. In diseases or injury of the blood vessels, there is the ability to use natural or synthetic vascular grafts [1]. After many years of experimental and clinical studies, it was stated that prostheses from polyethylene terephthalate and polytetrafluoroethylene are nearest to the model of the “ideal” vascular prosthesis [2,3,4]. Replacing pathologically changed blood vessels with polymer material can be a cause of disturbances in the blood haemostatic system. The surface of vascular implants always, to some degree, is subjected to interaction with blood platelets, and it influences the factors of the coagulation system and fibrinolysis. The result of that action is thrombus formation, which can lead, in the case of vascular prostheses, to their occlusion, or it can be a source of thrombotic–embolic complications [4,5,6].

A decrease in the ability of blood interaction with the surface of vascular prostheses can be achieved by changing their physicochemical properties as well as by introducing biologically active materials. The change of physical properties of knit prostheses of vascular vessels, such as porosity, knitting density, walls thickness, elasticity, longitudinal and transverse resistance to disruption, multidirectional resistance, susceptibility to dilatation and usable length, can be achieved by the use of proper textile technology and proper raw materials [7,8,9,10]. Modification of polyester knit by a change in the wettability degree of the surface is another way of decreasing the vascular prostheses thrombogenicity [11,12]. The wettability degree of the surface of vascular prostheses also influences their intrasurgical tightness; prostheses with hydrophilic surface characterized by a larger intrasurgical tightness in comparison with prostheses of hydrophobic surface [10,13,14].

Surface modification of vascular prostheses can be achieved, e.g., through placing prostheses in:alcoholic solution of a surface-active substance, e.g., Tween 80 and drying or,methanol solution and exposing them to the action of ultrasounds [10].

The prostheses of vascular vessels evaluated in the study were worked out on the basis of the original and modified procedure described in the text of the patent [14].

In the research on the improvement of the prosthesis features, it is important to learn about the haemostatic properties of the new blood vessel prostheses, as well as to determine the interaction with plasma factors of coagulation and fibrinolysis.

An evaluation of the influence of a knit hydrophilic prosthesis of blood vessels on the activation of the coagulation system in tests in vitro was the aim of the work.

## 2. Materials and Methods

### 2.1. Materials

The knitted implants vascular prosthesis Dallon^®^ and Dallon^®^ H were used for research. The Dallon^®^ H knitted vascular prosthesis double velour is characterized by a hydrophilic surface and made of non-resorbable polyester (poly-(ethylene terephalate)) yarn [14]. Conversely, the Dallon^®^ prosthesis is built by hydrophobic polyester (poly-(ethylene terephalate)) yarn. Both materials have been manufactured by TRICOMED SA, Lodz, Poland, and were CE-signed. Vascular prosthesis Dallon^®^ and Dallon^®^ H are presented in Figure 1.

### 2.2. Physic-Chemical Characteristic

X-ray diffraction measurements were performed by a PANalytical X’Pert Pro X-ray (Malvern Panalytical Ltd., Malvern, UK) diffractometer equipped with Ni-filtered Cu *Kα*_1_ radiation (*Kα*_1_ = 1.54060 Å, *U* = 40 kV, *I* = 30 mA) in the *2*θ range of 10–60° at room temperature. The surface morphology and the elements mapping were detected by an FEI Nova NanoSEM 230 (FEI Company, Hillsboro, OR, USA) scanning electron microscopy equipped with an EDS spectrometer (EDAX GenesisXM4, FEI Nova NanoSEM 230).

Samples of the vascular prosthesis were disintegrated in microtome (1 mg), dried 1 h at 70 °C under reduced pressure. Then, they were mechanically blended with 300 mg of the KBr, and disks were prepared at the pressure of a hydraulic press of 8 atm. under reduced pressure. The disk was disintegrated, and 280 mg of powder was dried again at the conditions described above [15]. After that, the KBr-disk was composed again. FTIR spectra of investigated samples were recorded on an FTIR (Spectro-Lab, Warszawa, Poland) spectrometer. The spectra were recorded in the range of wavenumber of 4000–650 cm^−1^, at the distribution of 4 cm^−1^. The background of the spectra was subtracted based on the individual peaks and on the group of peaks, using either the computer software of the instrument or Peak Fit for Windows. The peak areas were calculated using Voight’s or Lorentz’s function.

### 2.3. Differential Scanning Calorimetry

The study was carried out using the Perkin-Elmer DSC-7 instrument with the below-listed parameters [16].

-atmosphere–nitrogen, 30 cm^3^/min;-vessel–standard aluminium lid;-warming velocity—20 °C/min;-cooling velocity—20 °C/min;-range of temperature: 32–197 °C;-weight of sample: 6–10 mg.

The crystallinity of the samples (XDSC) was obtained from the differential scanning calorimetry (DSC) method, and it was calculated based on the following Equation (1):(1)$$X_{DSC}=\frac{\Delta H_{m}-\Delta H_{c}}{\Delta H_{ref}}\times 100\%$$
where:
Δ*H**_m_*—melting enthalpy;Δ*H**_c_*—crystallization enthalpy;Δ*H**_ref_*—refers to 100% crystalline polymer which in the case of poly-(ethyleneterephalate) equals to 140 ± 20 J/g.

### 2.4. Scanning Electron Microscopy with Energy Dispersive X-ray Spectroscopy

Morphology and chemical analysis of the samples were determined by using a Field Emission Scanning Electron Microscope (FEI Nova Nano SEM 230, FEI Company as a part of Thermo Fisher Scentific, FEI Company, Hillsboro, OR, USA) equipped with an Apollo X Silicon Drift Detector (SDD) together with EDAX Genesis Software (Mahwah, NJ, USA). The prepared samples were placed on the carbon stub and coated with a thin layer of gold (by using Auto 306 Coater with Turbomolecular Pumping System, Edwards Vacuum, West Sussex, UK) in order to obtain high-quality SEM images. SEM images and EDS measurements were carried out with an acceleration voltage of 5.0 and 10.0 kV, respectively. Moreover, SEM images were recorded in a beam deceleration mode to show more detailed features of the samples.

### 2.5. Coagulation System Studies

The studies were performed on human blood B Rh+ collected from an honorary donor, drawn as a single sterile PL-146 container (Baxter, Deerfield, IL, USA) and taken for preserving fluid CPD-A1 (citrate, glucose phosphate, adenine solution). Consent of the Bioethical Commission of Wroclaw Medical University was granted for the tests (No. KB-118/2011).

The prostheses samples with the mean weight of 0.4 g were placed in sterile polystyrene tubes, and 8.0 mL of blood citrate was measured into each of them [17,18,19,20,21,22,23]. Simultaneously with the tested samples, the tubes containing only blood were prepared as control samples. The tubes were placed on a haematological mixer, and they were put in motion (60 turns/min). The tests were performed at room temperature. The quantitative changes of blood coagulation parameters were determined after 30, 60, 90 and 120 min The whole blood after temporal contact with the samples of vascular prostheses and the blood constituting control was rotated 3000 rpm for 10 min, and next, plasma was separated from the mass of morphotic elements of blood. Mean-platelet-rich plasma was obtained. Activated partial thromboplastin time (APTT) and prothrombin time (PT) were determined in the plasma, and the prothrombin ratio (INR) was determined in the plasma. Further determination concerned: thrombin time (TT), fibrinogen concentration (Fb) and antithrombin III (AT III) activity, protein C (Prot C) activity, plasminogen (Plg) activity and factors XII, IX and VIII [20,21,22]. The tests of plasma were performed on the ACL 9000 analyser (Budapest, Hungary) at the temperature of (37 ± 1) °C, with the wavelength of 405 nm at a closed reagent system produced by Instrumentation Laboratory Diagnostic (Bedford, MA, USA). The determination procedures agree with the methods contained in the instruction of the device [24]. The results of the studies were subjected to statistical analysis with the use of the program Statistica 8.0. The arithmetical mean value and standard deviation were calculated. The essential differences in mean values were determined with *t*-test for independent tests. The correlation ratios were assumed to be important at *p* ≤ 0.05.

## 3. Results

### 3.1. Physic-Chemical Characteristic

The X-ray diffraction patterns of hydrophobic (Dallon^®^) and hydrophilic (Dallon^®^ H) prostheses are presented in Figure 2. Figure 2 shows numerous broad reflections associated with the semi-crystalline structure of both materials. The broad band located in the range of 15–27° is related to the amorphous halo and represented the amorphous structure of the polymer.

The Fourier transformed infrared spectra of testes materials are shown in Figure 3. In general, no differences between the materials are observed. The typical vibrational modes of poly (ethylene terephthalate) are described. No significant differences in the presence of additional absorption bands differed from the standard polyester FTIR spectrum were found for the unsealed vascular prostheses. The ratio of absorption band at =3450 cm^−1^ (O–H stretch) 1410 cm^−1^ (reference band) shows the increase for Dallon^®^ H (1.66) as compared with Dallon^®^ vascular grafts (1.44). This phenomenon is explained by the presence of a higher amount of hydroxyl groups on the Dallon H vascular prosthesis due to its modification process (Figure 3). Assignment of selected absorption bands of FTIR spectra of Dallon^®^ H and Dallon^®^ vascular prostheses are presented in Table 1.

### 3.2. Differential Scanning Calorimetry

The overlay diagram of DSC thermograms of Dallon^®^ and Dallon^®^ H vascular prostheses in the temperature range of 32–280 °C; an increase and decrease in temperature is shown in Figure 4. The diagrams of DSC both for Dallon^®^ or Dallon^®^ H vascular prostheses during an increase and decrease in temperature did not show any differences. A similar melting temperature (259 °C or 260 °C) as well as a similar crystallization temperature were found. The phenomenon above confirms the presence of a similar polymer in both studied vascular prostheses. The DSC results for Dallon^®^ H or Dallon^®^ vascular prostheses are shown in Table 2.

### 3.3. Scanning Electron Microscopy with Energy Dispersive X-ray Spectroscopy

Scanning electron microscopy (SEM) images of the Dallon^®^ and Dallon^®^ H vascular prosthesis samples are presented in Figure 5. It can be seen that both knitted prosthesis shows parallel and uniform randomly oriented fibres with relatively smooth surfaces. No damages or fibre cracks in the samples were also observed. The average fibre diameter in the samples was (15 ± 2) μm.

The next step of Scanning electron microscopy with energy dispersive X-ray spectroscopy (SEM-EDS) investigation was the analysis of the elemental mapping, which was performed in order to reveal the distribution of elements in the samples. The EDS mapping of knitted prosthesis surfaces (see Figure 6) shows that carbon (red colour) and oxygen (green colour) are uniformly distributed.

### 3.4. Coagulation System Study

Biological tests in vitro aimed at studying quantitative changes of the chosen coagulation parameters after temporal contact of blood with a Dallon^®^ H vascular prosthesis were performed. The evaluation of those changes was used for the determination of the activation of the plasmatic coagulation system. The activation of the coagulation system dependent on the contact factors (endogenous system) was evaluated with an Activated partial thromboplastin time (APTT) test and through the determination of the activity of factors: FXII, FIX and FVIII. The activation of coagulation dependent on tissue thromboplastin (exogenous system) was evaluated with a Prothrombin time (PT) test. The Thrombin time (TT) test was the measurement connecting both coagulation systems. The TT test states the conversion of fibrinogen into fibrin and depends on the amount of fibrinogen, the concentration of which was also determined. The activation of the fibrinolytic system was evaluated through the determination of plasminogen activity. Antithrombin III and protein C activity were determined from coagulation inhibitors [20,21,22,23]. The reference values of coagulation parameters for human plasma determined on the ACL 9000 analyser are presented in Table 3.

Mean coagulation system parameter values with control plasma and studies with standard deviation as well as the level of significance are given in Figure 7, Figure 8, Figure 9, Figure 10, Figure 11, Figure 12, Figure 13, Figure 14, Figure 15, Figure 16 and Figure 17. After blood contact with Dallon^®^ H vascular prosthesis in the plasma, an important lengthening of a APTT value by 6% (*p* < 0.01) and11% (*p* < 0.001) after 30 and 60 min and 13% (*p* < 0.001) after 90–120 min, respectively, was observed in relation to the control value. The lengthening of the APTT showed an important difference (*p* < 0.001) in relation to the primary value (time 0). The biggest lengthening, by an average of 13%, was observed in 90–120 min (Figure 7). Factor XII (FXII) activity in the plasma had an important decrease by 5% (*p* < 0.01) and 3% (*p* < 0.05) after 30 and 60 min in relation to the control group. After 90 and 120 min, F XII decreased activity by 6% (*p* < 0.001) in relation to the control and primary value (Figure 8). Factor IX (FIX) activity had an important decrease by 8% (*p* < 0.01) after 30, 60 and 90 min and by 14% (*p* < 0.01) after 120 min in relation to the control group. Factor IX activity had an important decrease by 12% (*p* < 0.01, *p* < 0.001) after 30 and 60 min and 17% (*p* < 0.01) after 90 and 120 min compared to the primary value (Figure 9). Factor VIII activity had an important decrease by 6% (*p* < 0.05) after 30 and 60 min and by 10% (*p* < 0.01) after 90 min in relation to the control value. After 90 min, at 120 min, an important decrease by 11% (*p* < 0.05) was observed in relation to the primary value (Figure 10). The prothrombin time and prothrombin ratio in the plasma in all measurement periods were within the control group and primary values (Figure 11 and Figure 12). The thrombin time (TT) had important lengthening by 3% (*p* < 0.05) after 30 min and 60 min in relation to the control group value. TT values obtained in the measurement period of 90–120 min compared to the primary value can be seen in Figure 13. The fibrinogen concentration in the plasma had an important decrease by 8% (*p* < 0.01) after 30 min and by 3% (*p* < 0.05) after 60 min in relation to the control value. Fibrinogen values had an important decrease by 5% (*p* < 0.05) after 30 min in relation to the primary value (Figure 14). Plasminogen activity in all measurement periods showed no important difference compared to the control group value and was close to the primary value (Figure 15). Antithrombin III activity and protein C activity in the plasma in all measurement periods and in the control group was close to the primary value. The obtained values in those observation periods were at comparable levels (Figure 16 and Figure 17).

## 4. Discussion

Fourier transform infrared spectroscopy (FTIR) analysis of Dallon^®^ H modified vascular prosthesis confirms the effectiveness of the implant modification process, which resulted in the hydrophilic behaviour. One difference in FTIR spectra of Dallon^®^ and Dallon^®^ H vascular prosthesis is higher absorption at the wavenumber of 3450 cm^−1^ responsible for the presence of higher numbers of hydroxyl groups as well as at the wavenumber of 1630 cm^−1^ directly connected with the presence of free carboxylic groups. DSC study allowed for the estimation of the surface vascular prosthesis modification effect on the polymer crystallinity. The crystallinity is a very important factor affecting the long-term implant resistance on the degradation after grafting. As shown in [25,26], the knitted vascular prostheses left their mechanical resistance after implantation. The degradation process resulted in hydrolysis of amorphous regions and further crystalline regions, and this process is the most significant factor that influences the above-mentioned phenomenon. DSC studies confirmed the insignificant reduction in crystallinity index after the modification process of Dallon^®^ H vascular prostheses in comparison to Dallon^®^ vascular prostheses. The surface modification of vascular prostheses, yielding an increase in hydrophilic behaviour, affects the facility of preclotting: the blood demand is 5–8 times lower and the “squeezing” and “massaging” of the prosthesis before implantation is eliminated [13,20,27].

Biological tests in vitro were conducted to studying quantitative changes of the chosen coagulation parameters after temporal contact of blood with a vascular prosthesis Dallon^®^ H and evaluation of those changes was used for determination of the activation of the plasmatic coagulation system [21,22]. The result of the coagulation study is the formation of a thrombi-wall of varying thicknesses and sizes. The establishment of blood clots in the pores on the surface of the prosthesis after the introduction into the circular system is positive for the mechanism of sealing and healing of the vascular graft. Excessive severity of the process and the associated oxidative stress may lead to narrowing or blockage of the prosthesis [28]. The state of haemostatic balance is determined mainly by the inner layer of the prosthesis, which is in contact with flowing blood. Both the thickness of the inner layer and preservation of its patency depends on the mutual relationship between blood components having procoagulant and anticoagulant action in the early period after vascular graft implantation [10,19,21,26,28]. It is known that biomaterials in contact with the circulatory system can affect the coagulation system [29,30]. The influence of vascular prostheses on microplate fractions and activation of platelets under static and dynamic conditions is presented in the article by M.H. Struszczyk [10]. The authors showed a slight effect of the Dalton prosthesis on the fraction of microplates and platelets under static and dynamic conditions and on the activation of platelet aggregates. An increased number of platelets and their aggregates in a dynamic and flow system was demonstrated.

The study conducted by Paluch and Szymonowicz [20] presents studies of the effect of polyester knitted fabrics with various levels of surface wettability on the coagulation system, fibrinolysis system as well as on haematological parameters of blood in in vitro tests. The tests were carried out on whole blood, collected for an anticoagulant, which did not clot under dynamic conditions. Changes in the red blood cell system were assessed by the determination of haematocrit (Ht), haemoglobin (Hb), red blood cell (RBC) and red blood cell count: mean red blood cell volume (MCV), mean red blood haemoglobin mass (MCH) and mean red blood cell concentration and haemoglobin in red blood cells (MCHC). Changes in the white blood cell system were assessed by determining the number of white blood cells (WBC), taking into account the leukogram: percentage of granulocytes (GRA), lymphocytes (LYM) and monocytes (MON). Assessment of platelets was based on the determination of beta thromboglobulin (beta-TG) levels and the determination of platelets (PLT), mean platelet volume (MPV) and platelet haematocrit (PCT). The authors found that temporary, direct contact of blood with knitted fabrics with hydrophobic and hydrophilic surfaces does not affect the quantitative and morphological changes of red blood cells, but it affects the morphology and the number of platelets and white blood cells. These changes are related not only to the composition, structure and surface properties of the tested materials but also to the metabolic properties of white blood cells and platelets. After the contact of blood with the hydrophilic fabric, changes in the number of platelets appeared in the initial period of observation. This is probably related to the process of rapid soaking of the material with blood, which causes, to a greater extent than to the hydrophobic surface, protein adsorption, then adhesion of platelets and their activation (increase in beta-TG) and interaction with leukocytes. The morphological assessment of the RBC, WBC and PLT was made on the basis of the topography of knitted fabrics in a scanning electron microscope.

Scanning microscopy showed the adhesion of PLT, RBC and WBC to the surface of prostheses [20]. In the scanning microscopic image, no differences in the morphology of platelets were found after contact with knitted fabrics with hydrophobic and hydrophilic surfaces. Platelets are both spherical and highly flattened with numerous pseudopodia of various lengths that increase their adhesion to the surface and mutual contact. The platelets are connected to each other and form smaller or larger aggregates. The change in the shape of blood cells indicates their stimulation and activation.

Our research on vascular prostheses was a supplement to the assessment only by activating the plasma coagulation system because this process is one of the most important in sealing the prosthesis in order to check whether the obtained values are comparable to those obtained for the knitted fabrics presented in the article by Paluch and Szymonowicz, which are the starting material for the production of Dallon^®^ and Dallon^®^ H prostheses [20].

In the performed studies in vitro of a prosthesis Dallon^®^ H in contact with blood, changes in the parameters of endogenous coagulation system (APTT, FXII and FIX) were observed without changes in the exogenous system (PT). The observed changes in the values of APTT, FXII, FIX, FVIII, TT and Fb did not exceed the referential values for these indexes (Table 3).

In plasma after contact with a Dallon^®^ H vascular prosthesis, the prolongation of APTT and a decrease in the activity of factors XII, IX and VIII were observed. The largest changes were observed after 90 min of contact of blood with the material. Further prolongation of the test’s time did not essentially influence the value change of those indexes. Values PT at all times of observation were comparable with the value in the control group. The prolongation of the TT time and a decrease in fibrinogen concentration were observed until 60 min. After 90 and 120 min, the observed values approached the control group and on an equivalent level. The activity of the natural coagulation anticoagulants, ATIII and protein C, remained unchanged. The activity of plasminogen at all times of the measurement was on the level of control value. The observed values were the result of absorption of blood through the vascular graft and its quick sealing. That probably caused the wear and inactivation of components of the plasma coagulation system. The sealing process in the hydrophilic Dallon^®^ H after entering the bloodstream will be faster and with less blood loss. Hypercoagulability in the early period post-implantation is a favourable phenomenon, which allows quick sealing of the Dallon^®^ H [28]. The activation of the coagulation system dependent on contact factors may invoke intravascular coagulation activity, but with the undisturbed functioning of the organism, correct haemopoiesis of blood and the correct efficient fibrinolysis natural coagulation inhibitor level, thrombus formation should be slight, especially in the initial period of the healing of the vascular prosthesis.

A significant impact of surface wettability on reducing the activation of the clotting processes showed the in vitro tests carried out by Paluch and Szymonowicz [12,20] that indicate an extended clotting time and faster blood saturation in the case of the polyester hydrophilic knitted compared to the hydrophobic [12,20]. The paper by Paluch and Szymonowicz [20] presents the effect of polyester knitted fabrics on the coagulation process, fibrinolysis after temporary contact with whole blood after 5, 15, 30, 60, 90, 120, 180 and 240 min. Polyester knitted fabric with a hydrophobic surface did not cause significant changes in the determined parameters of the coagulation system and the fibrinolytic system. On the other hand, the polyester knitted fabric with a hydrophilic surface causes a significant extension of the clotting time and a decrease in the activity of factor XII, IX and VIII, and prolonged aPTT compared to the polyester knitted fabric with a hydrophobic surface. Extending the study time to 240 min had no significant effect on the values of these parameters, and the greatest changes were observed after 90 min with contact between blood and the material. The values of PT and TT, as well as the concentration of fibrinogen at all observation times, are comparable with the value in the control group and in the group after contact with the hydrophobic knitted fabric. The activity of the natural coagulation anticoagulants, antithrombin III and protein C and the activity of plasminogen also remained unchanged. The studies showed that the knitted fabric with a hydrophilic surface prolongs the clotting time in the intrinsic system and reduces the activity of the factors of this system without affecting the extrinsic coagulation [20]. The research carried out by Paluch and Szymonowicz [20] allows for the conclusion that due to its properties, the hydrophilic surface of the knitted fabric absorbs blood faster than the hydrophobic fabric, causes greater changes in the blood picture. The process of sealing the hydrophilic knit once it enters the bloodstream will be faster and with less blood loss. Activation of the coagulation system dependent on contact factors may activate intravascular coagulation, but with undisturbed body functions, normal hemopoiesis of blood morphotic elements and synthesis of plasma coagulation factors, efficiently functioning fibrinolysis and the level of natural coagulation inhibitors, the process of thrombus formation should be minimal, and it is in the initial period of the knitted fabric healing.

Based on the previously conducted research [20] of polyester knitted fabrics with a hydrophilic surface, for which the greatest changes were found between 30 and 120 min, the Dallon^®^ H vascular prosthesis was assessed in this respect. The values of the coagulation system parameters obtained in our research after this contact time are comparable with the values obtained for hydrophilic polyester knitted fabrics.

This is associated with the beneficial properties of minimizing the process of clotting and thus reducing the amount of blood needed to saturate material during implantation. Implantation and intraoperative studies of the Dallon^®^ and Dallon^®^ H vascular prostheses for the porcine thoracic aorta, carried out by Milewski et al. [13] and Paluch and Szymonowicz [27], showed that preclotting is not necessary in the case of the Dallon^®^ H, and the healing process of both prostheses is similar and correct [13,28]. Moreover, published studies by Karaszewska and Buchenska [31,32] showed the possibility of an additional modification of the Dallon^®^ H by saturation with amoxicillin, which additionally increases its properties with antibacterial activity in relation to strains responsible for hospital infections, such as *S. aureus* and *E. coli*. In addition, the researchers observed that saturation of the Dallon^®^ H vascular prosthesis with biocide did not cause irritation in in vivo studies carried out on rabbits [32]. The above properties of these prostheses indicate that the Dallon^®^ H is a material with significant clinical potential, especially considering the fact that the introduction of knitted vascular prostheses to the bloodstream may cause an imbalance in the haemostatic system.

## 5. Conclusions

Vascular prosthesis Dallon^®^ H bilateral velvet is characterized by hydrophilic properties and large mechanical strength. In the studies of the plasmatic coagulation system, after a temporal contact of blood with the Dallon^®^ H, the prolongation of partial thromboplastin time after the activation and decreased activity of factor XII, IX and VIII with the normal prothrombin time values were observed. Prolongation of thrombin time and decrease in fibrinogen concentration was observed in the initial period of the experiment. The activity of plasminogen and natural anticoagulants of coagulation, antithrombin III and protein C remained unchanged. Vascular prosthesis Dallon^®^ H in contact with blood affects the coagulation parameters, which is the result of absorption of blood through the prosthesis and its seal.

## Figures and Tables

**Figure 1 nanomaterials-11-01600-f001:**
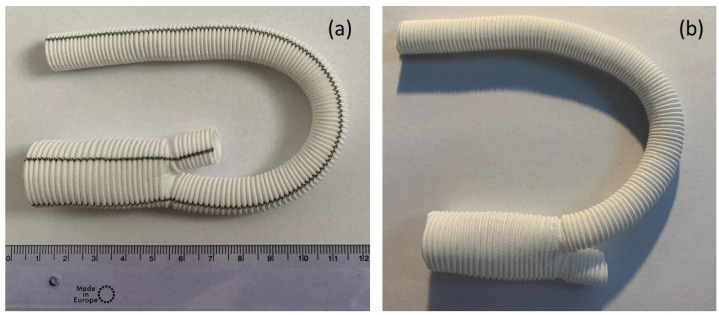
The knitted implants vascular prosthesis Dallon^®^ (**a**) and Dallon^®^ H (**b**).

**Figure 2 nanomaterials-11-01600-f002:**
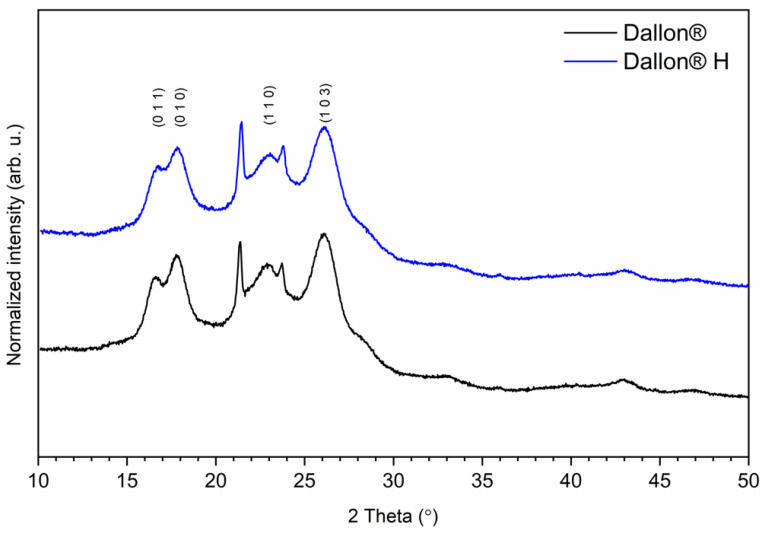
X-ray powder diffraction pattern of Dallon^®^ and Dallon^®^ H vascular prostheses.

**Figure 3 nanomaterials-11-01600-f003:**
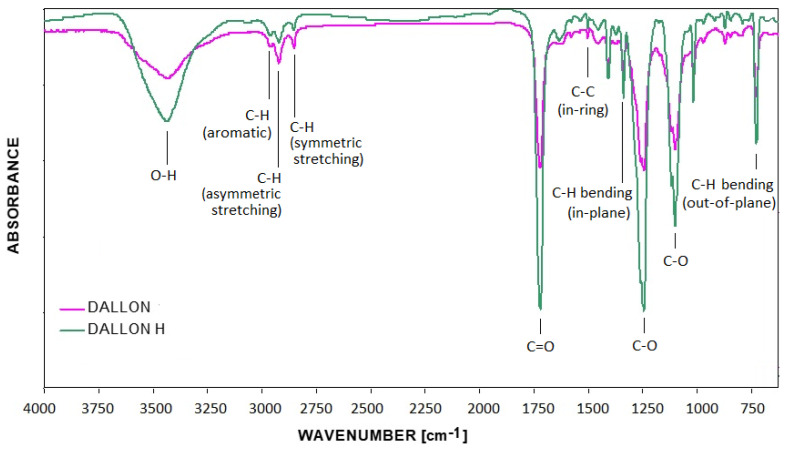
FTIR spectra of Dallon^®^ and Dallon^®^ H vascular prostheses made of un-modified polyester poly (ethylene terephalate) yarn.

**Figure 4 nanomaterials-11-01600-f004:**
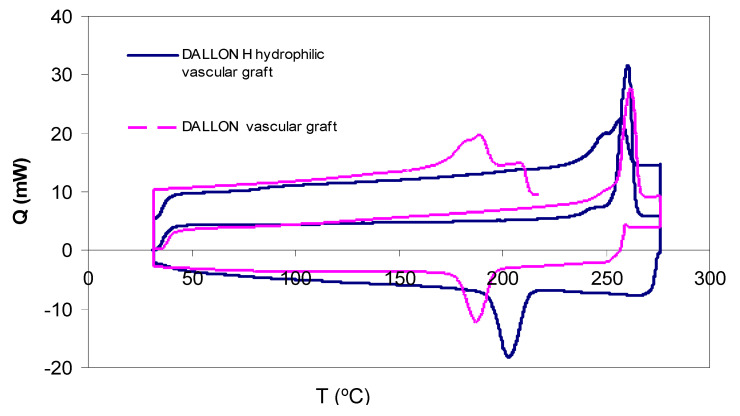
Differential scanning calorimetry data for Dallon^®^ H and Dallon^®^ vascular prostheses made of un-modified polyester poly (ethylene terephalate) yarn.

**Figure 5 nanomaterials-11-01600-f005:**
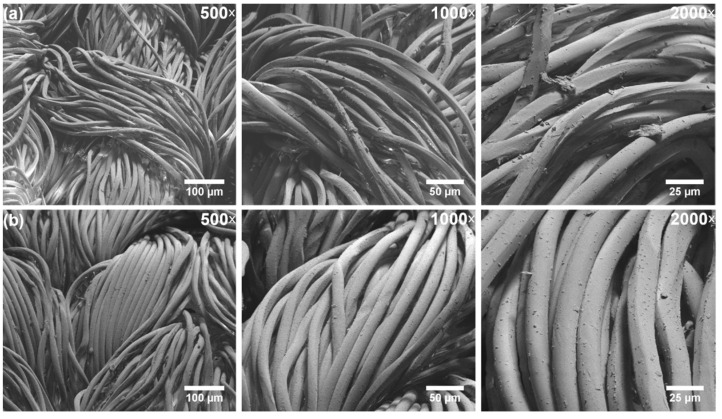
Representative low magnification SEM images of the (**a**) Dallon^®^ and (**b**) Dallon^®^ H vascular prostheses recorded for 500-, 1000- and 2000-times magnification, respectively.

**Figure 6 nanomaterials-11-01600-f006:**
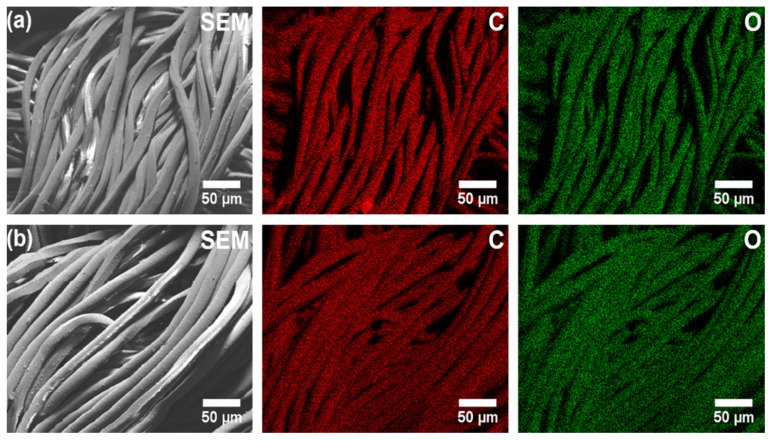
Elemental mapping distribution (magnification of 1000× were used for each image) of carbon (C) and oxygen (O) on the surface of the (**a**) Dallon^®^ and (**b**) Dallon^®^ H vascular prostheses samples.

**Figure 7 nanomaterials-11-01600-f007:**
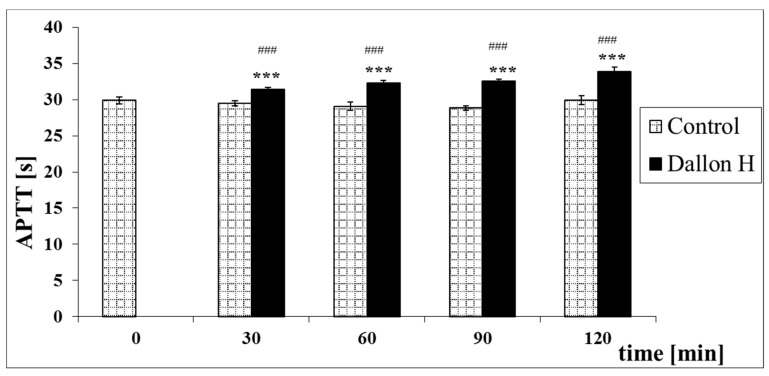
Activated partial thromboplastin time (APTT). *** *p* < 0.001 to the control, ### *p* < 0.001 to the primary value.

**Figure 8 nanomaterials-11-01600-f008:**
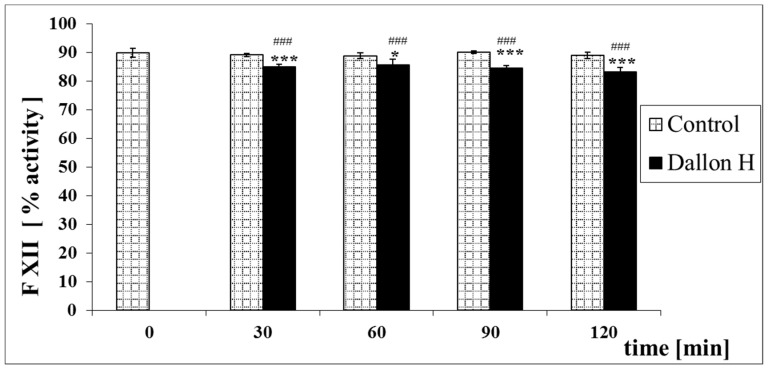
Factor XII (FXII) activity. * *p* < 0.05, *** *p* < 0.001 to the control group. ### *p* < 0.001 to the primary value.

**Figure 9 nanomaterials-11-01600-f009:**
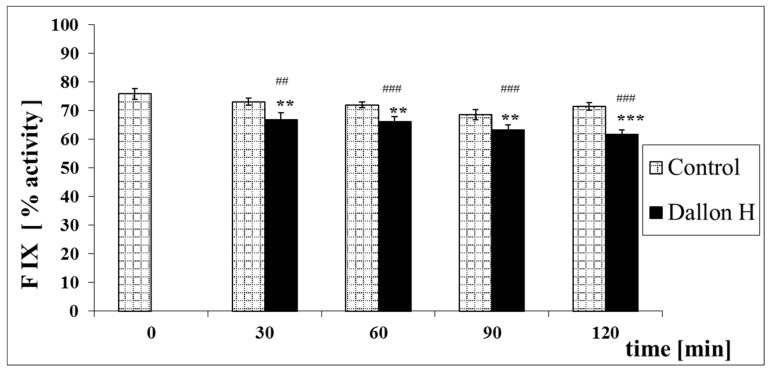
Factor IX (FIX) activity *p* < 0.01, ** *p* < 0.01, *** *p* < 0.001 to the control group; ## *p* < 0.01, ### *p* < 0.001 to the primary value.

**Figure 10 nanomaterials-11-01600-f010:**
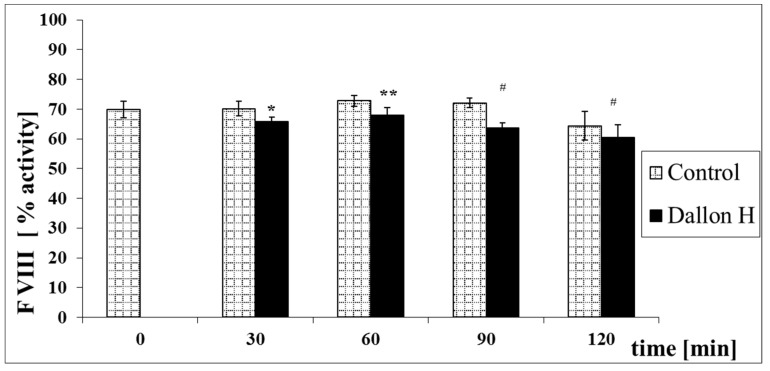
Factor VIII (FVIII) activity. * *p* < 0.05, ** *p* < 0.01 to the control group. # *p* < 0.05 to the primary value.

**Figure 11 nanomaterials-11-01600-f011:**
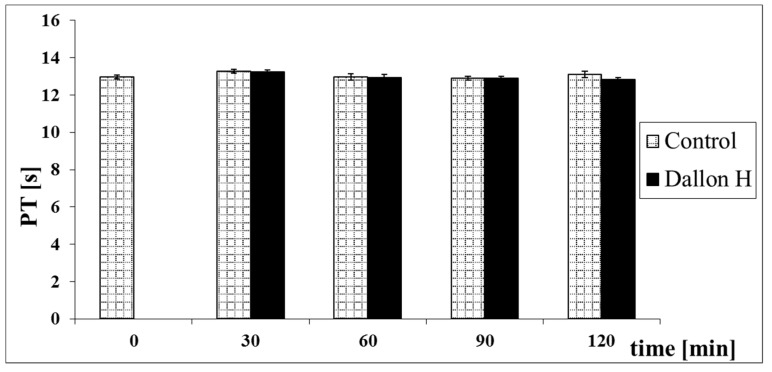
Prothrombin time (PT).

**Figure 12 nanomaterials-11-01600-f012:**
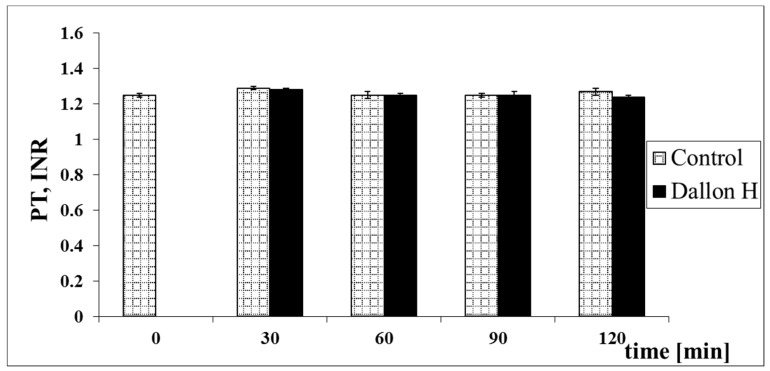
Prothrombin ratio (INR).

**Figure 13 nanomaterials-11-01600-f013:**
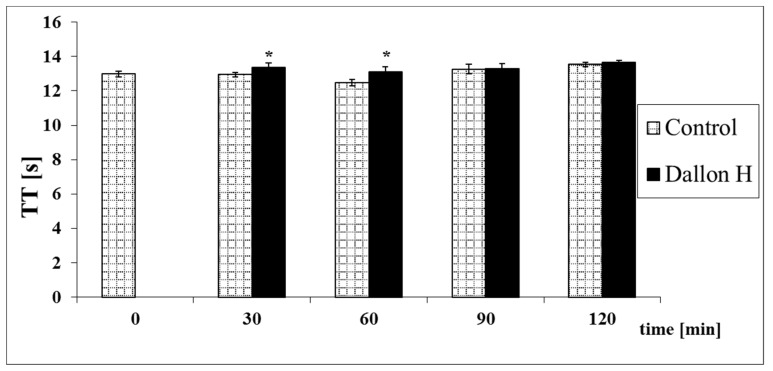
Thrombin time (TT). * *p* < 0.05 to the control group.

**Figure 14 nanomaterials-11-01600-f014:**
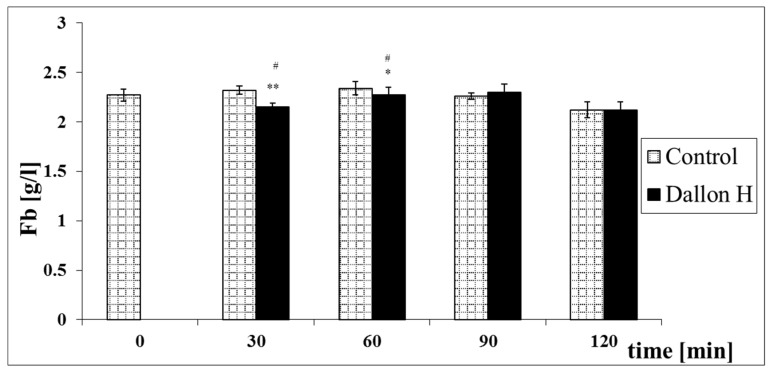
Fibrinogen concentration (Fb). * *p* < 0.05, ** *p* < 0.01 to the control. # *p* < 0.05 to the primary value.

**Figure 15 nanomaterials-11-01600-f015:**
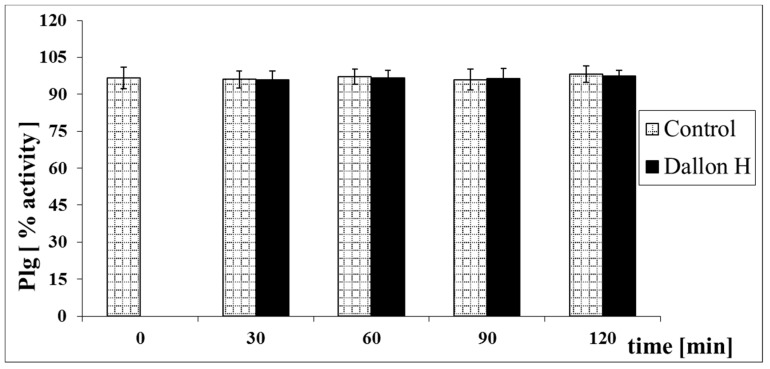
Plasminogen (Plg) activity.

**Figure 16 nanomaterials-11-01600-f016:**
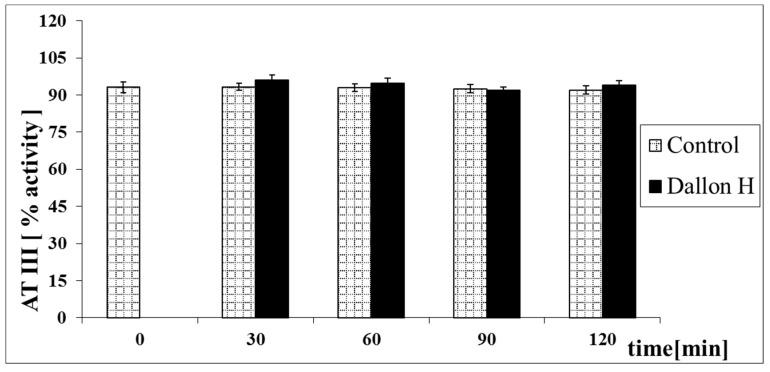
Antithrombin III (AT III) activity.

**Figure 17 nanomaterials-11-01600-f017:**
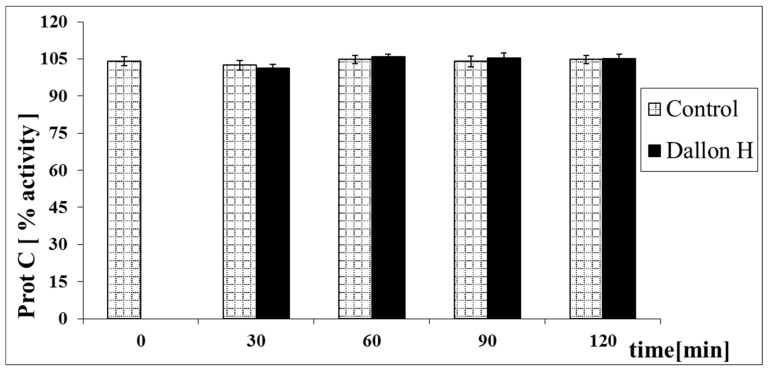
Protein C (Prot C) activity.

**Table 1 nanomaterials-11-01600-t001:** Assignment of selected absorption bands of FTIR spectra of Dallon^®^ H vascular prosthesis.

Lp.	Wavelength [cm^−1^]	Suggested Infrared Band Assignment
1	3445	O–H stretching bond
2	3050	Aromatic C–H stretch
3	3000–2800	Stretching C–H bond
4	1720, 1630	C=O
5	1580, 1500	C–C stretch (in-ring)
6	1310	C–H bending (in-plane)
7	730	C–H bending (out-of-plane)
8	1260, 1100	C–O bond from ester group
9	973	Trans configuration (crystalline band)
10	927	Cis configuration
11	900	Gauche configuration

**Table 2 nanomaterials-11-01600-t002:** The results of DSC study of Dallon^®^ or Dallon^®^ H vascular prosthesis.

ΔCp	Melting Temperature	Melting Enthalpy	Crystallization Temperature	Crystallization Enthalpy	Reference Enthalpy	Crystallinity
[J/g K]	Tm [C]	Δ*H**_m_*	Tc [C]	Δ*H**_c_*	Δ*H**_ref_*	*X_DSC_*
Dallon^®^						
0.12	259	64.9	207	43.5	140	0.15
Dallon^®^ H						
0.14	260	62.6	207	46.6	140	0.11

**Table 3 nanomaterials-11-01600-t003:** Reference values parameters of the coagulation system.

Parameters	Reference Values Range
APTT	25.0–37.0 s
PT	9.0–12.6 s
PT INR	0.8–1.2
Fb	2.0–4.5 g/L
TT	11.0–18.0 s
Factor XII	50–150% activity
Factor IX	65–150% activity
Factor VIII	50–150% activity
AT III	75–122% activity
Protein C	6–134% activity
Plazminogen	75–150% activity

## Data Availability

Not applicable.

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
