# Peer review of "The Influence of a Knitted Hydrophilic Prosthesis of Blood Vessels on the Activation of Coagulation System—In Vitro Study"

_nanomaterials, 2021, doi:10.3390/nano11061600_

Round 1

Reviewer 1 Report

The present manuscript entitled “The Influence of a Knitted Hydrophilic Prosthesis of Blood Vessels on the Activation of Coagulation System - In Vitro 3 Study” co-authored by Szymonowicz et al., deals with the improvement of the vascular prosthesis features, while playing with the physico-chemical properties of the devices. A special attention is paid on the hydrophobic/hydrophylic properties of the final materials, as well as their impact on the coagulation blood system activation.

Overall, the study presents an interesting problematic of common clinical importance, relative to the blood coagulation properties as a response of the blood contact with the vascular prosthesis.

After carefully investigating the study, I might have several major remarks:

1. Did the authors studied the blood cell (platelet, erythrocyte) adhesion on the vascular prosthesis surface? It will be very interesting to demonstrate the absence/presence of cloth formation (fibrinogen network formation) on the device surface and interpret the data in respect to the hydrophobic/hydrophylic nature of the device. Please comment with the available scientific literature. Is the porous structure (pores between the individual fibers) of the proposed Dallon devices, present a risk for the blood penetration and diffusion through the samples wall?

2. Is the 'wave' like structure of the device is impacting the circulation of the blood ? A reference presenting uniform surface will be of advantage in this study.

3. Concerning the wettability of the final devices, the Reviewer strong recommendation is to present relative contact angle studies to Dallon(R)H and Dallon (R) and discuss the obtained data, while taking into account Comment 1 described above. Chemical structure of the used polymers for the production of the Dallon(R)/H, will be helpful.  

4. The Reviewer suggestion is to remove equation (1) and (2) from the experimental part, as this information is commonly used in the data treatment software.

5. What is the blank used in the FT-IR analysis, H2O/CO2? Could you please comment the characteristic band around 1650 cm-1 that is only present in the Dallon(R)H spectum?

6. Please correct line 188 "...DSC curves .." , to "...DSC thermograms...". Please clarify line 206 "...randomly oriented...", or detail the used knit pattern. 

7. Figure 4: please add "degrees" in the X-axis title. The Reviewer suggestion is to delete Figure 5, as it contains the same information already described in Table 2.

8. The authors mention that the "The crystallinity is very important factor effecting the long-term implant resistance on the degradation after grafting. As shown in [25, 26] the knitted vascular prostheses left their mechanical resistance after implantation.'' (line 275-277). In this meaning, did the authors performed mechanical tests on the discussed Dallon-based samples? Please compare with the mechanical properties of other devices used in the field.

Based on the comments presented above, the Reviewer recommend to carefully rework the manuscript, with respect to the described major revision remarks presented above, before considering the study for publication.

Author Response

Dear Editor,

We would like to express our sincerest gratitude to the Reviewers for their enormous efforts in criticizing the manuscript. We have taken into account all raised question here follows the detailed answers to the Reviewers. Moreover, all changes we have made to the original manuscript, are marked in the red colour in the text.

Reviewer 1:

The present manuscript entitled “The Influence of a Knitted Hydrophilic Prosthesis of Blood Vessels on the Activation of Coagulation System - In Vitro 3 Study” co-authored by Szymonowicz et al., deals with the improvement of the vascular prosthesis features, while playing with the physico-chemical properties of the devices. A special attention is paid on the hydrophobic/hydrophylic properties of the final materials, as well as their impact on the coagulation blood system activation.

Overall, the study presents an interesting problematic of common clinical importance, relative to the blood coagulation properties as a response of the blood contact with the vascular prosthesis.

After carefully investigating the study, I might have several major remarks:

Q1. Did the authors studied the blood cell (platelet, erythrocyte) adhesion on the vascular prosthesis surface? It will be very interesting to demonstrate the absence/presence of cloth formation (fibrinogen network formation) on the device surface and interpret the data in respect to the hydrophobic/hydrophylic nature of the device. Please comment with the available scientific literature.

Answer: Thank you for the suggestion. It was added these significant points to the discussion. It is known that biomaterials in contact with the circulatory system can affect the coagulation system (Grunkemeier, Goodman). The influence of vascular prostheses on microplate fractions and activation of platelets under static and dynamic conditions is presented in the article by M.H. Struszczyk [10]. The authors showed a slight effect of the Dalton prosthesis on the fraction of microplates and platelets under static and dynamic conditions and on the activation of platelet aggregates. An increased number of platelets and their aggregates in a dynamic and flow system was demonstrated.

The study conducted by Paluch and Szymonowicz [20] presents studies of the effect of polyester knitted fabrics with various levels of surface wettability on the coagulation system, fibrinolysis system as well as on hematological parameters of blood in in vitro tests. The tests were carried out on whole blood, collected for an anticoagulant, which did not clot under dynamic conditions. Changes in the red blood cell system were assessed by the determination of hematocrit (Ht), hemoglobin (Hb), red blood cell (RBC) and red blood cell counts: mean red blood cell volume (MCV), mean red blood hemoglobin mass (MCH) and mean red blood cell concentration and hemoglobin in red blood cells (MCHC). Changes in the white blood cell system were assessed by determining the number of white blood cells (WBC) taking into account the leukogram: percentage of granulocytes (GRA), lymphocytes (LYM) and monocytes (MON). Assessment of platelets was based on the determination of beta thromboglobulin (beta-TG) levels and the determination of platelets (PLT), mean platelet volume (MPV) and platelet hematocrit (PCT). The authors found that temporary, direct contact of blood with knitted fabrics with hydrophobic and hydrophilic surfaces does not affect the quantitative and morphological changes of red blood cells, but it affects the morphology and the number of platelets and white blood cells. These changes are related not only to the composition, structure and surface properties of the tested materials, but also to the metabolic properties of white blood cells and platelets. After the contact of blood with the hydrophilic fabric, changes in the number of platelets appeared in the initial period of observation. This is probably related to the process of rapid soaking of the material with blood, which causes, to a greater extent than to the hydrophobic surface, protein adsorption, then adhesion of platelets and their activation (increase in beta-TG) and interaction with leukocytes. The morphological assessment of the RBC, WBC, PLT was made on the basis of the topography of knitted fabrics in a scanning electron microscope.

Scanning microscopy showed the adhesion of PLT, RBC and WBC to the surface of prostheses [20]. In the scanning microscopic image, no differences in the morphology of platelets were found after contact with knitted fabrics with hydrophobic and hydrophilic surfaces. Platelets are both spherical and highly flattened with numerous pseudopodia of various lengths that increase their adhesion to the surface and mutual contact. The platelets are connected to each other and form smaller or larger aggregates. The change in the shape of blood cells indicates their stimulation and activation.

Our research on vascular prostheses was a supplement to the assessment only by activating the plasma coagulation system, because this process is one of the most important in sealing the prosthesis, in order to check whether the obtained values ​​are comparable to those obtained for the knitted fabrics presented in the article by Paluch and Szymonowicz, which are the starting material for the production of Dallon and Dallon H prostheses [20].

The research carried out by Paluch and Szymonowicz [20] allows for the conclusion that due to its properties, the hydrophilic surface of the knitted fabric absorbs blood faster than the hydrophobic fabric, causes greater changes in the blood picture. The process of sealing the hydrophilic knit once it enters the bloodstream will be faster and with less blood loss. Activation of the coagulation system dependent on contact factors may activate intravascular coagulation, but with undisturbed body functions, normal hemopoiesis of blood morphotic elements and synthesis of plasma coagulation factors, efficiently functioning fibrinolysis and the level of natural coagulation inhibitors, the process of thrombus formation should be minimal, and it is in the initial period of the knitted fabric healing.

Q1a. Is the porous structure (pores between the individual fibers) of the proposed Dallon devices, present a risk for the blood penetration and diffusion through the samples wall?

Answer: Dallon and Dallon H vasular prostheses are pre-clotted before implantation. This procedure involves the prosthesis being sitheed with the patient's blood, thus sealing the wall of the prosthesis. In the case of Dallon H, there is rapid preclotting, thanks to the modification of the surface increasing hydrophilia (blood consumption is 5 times lower) [Struszczyk 2002, Polymers]. The use of pre-clotting procedure is beneficial in the system of preoperative sealing of vascular prostheses, as the resulting, in this process, a thin wall-side clot seals the prosthesis and positively affects its healing in. In addition, the porosity and fabric structure of Dallon and Dallon H prostheses allows optimal tissue in-growth. [Toe 2001].

Q2. Is the 'wave' like structure of the device is impacting the circulation of the blood? A reference presenting uniform surface will be of advantage in this study.

Answer: Dallon and Dallon H vascular prostheses are designed by knitting technique, have a flexible and enhanced structure (double velour) that allows the prosthesis to fit into the patient's vascular system. Double velour structure of the prostheses (both,  Dallon being the reference and Dallon H) allows to prevent the blood platelets accumulation onto the internal prostheses by the formation of  the microturbulence that reduce their thrombogenicity.

Q3. Concerning the wettability of the final devices, the Reviewer strong recommendation is to present relative contact angle studies to Dallon(R)H and Dallon (R) and discuss the obtained data, while taking into account Comment 1 described above. Chemical structure of the used polymers for the production of the Dallon(R)/H, will be helpful.  

Answer: The relative contact angle of Dallon H prosthesis is not possible to determine to the high hydrophilic behaviour of its fabric surface. Water drops were immediately absorbed into the Dallon H prosthesis surface.

Q4. The Reviewer suggestion is to remove equation (1) and (2) from the experimental part, as this information is commonly used in the data treatment software.

Answer: The equation (1) was removed from article, however, due the reference to 100% crystalline polymer, we propose to leave the equation (2).

Q5. What is the blank used in the FT-IR analysis, H2O/CO2? Could you please comment the characteristic band around 1650 cm-1 that is only present in the Dallon(R)H spectum?

Answer: Due to the FTIR technique and background separation via computer software used we did not use the blank. As described in article we separated only absorption band at wavenumber ca 1630 cm-1, related to C=O.

Q6. Please correct line 188 "...DSC curves .." , to "...DSC thermograms...". Please clarify line 206 "...randomly oriented...", or detail the used knit pattern. 

Answer: The sentence in line 188 was corrected. We postulate to leave the sentence presented in line 206 due to the relation of the double velour knitted technique that randomly enhanced the internal and external surface of the vascular prosthesis: internally supported the reduction of the thrombogenicity and improvement of the neointima formation and fixation, externally improved the tissue ingrowth.

Q7. Figure 4: please add "degrees" in the X-axis title. The Reviewer suggestion is to delete Figure 5, as it contains the same information already described in Table 2.

Answer: The “degree” in Figure 4 was added and the Figure 5 was deleted.

Q8. The authors mention that the "The crystallinity is very important factor effecting the long-term implant resistance on the degradation after grafting. As shown in [25, 26] the knitted vascular prostheses left their mechanical resistance after implantation.'' (line 275-277). In this meaning, did the authors performed mechanical tests on the discussed Dallon-based samples? Please compare with the mechanical properties of other devices used in the field.

Answer: The mechanical behaviour (static and dynamic) of the disused vascular prostheses were tested acc. ISO 7198 and EN 12006-2 Standards. The testing covers several aspects of the determination: longitudinal tensile strength, bursting strength, relaxed inner diameter, pressurized inner diameter and suture retention strength. Range of above-mentioned testes requires additional publication. Generally, the hydrophilization of the vascular prosthesis did not affected the mechanical properties and all received results indicates the safety of the medical devices.    

Reviewer 2 Report

This manuscript reports the Influence of a knitted hydrophilic prosthesis of blood vessels on the activation of coagulation system. The experimental parts is clearly written as the entire paper. The paper is ready for publication in Nanomaterials.

Author Response

Dear Editor,

We would like to express our sincerest gratitude to the Reviewer for his/her enormous efforts in criticizing the manuscript and we would like to thank for such review.

Reviewer 2:

This manuscript reports the Influence of a knitted hydrophilic prosthesis of blood vessels on the activation of coagulation system. The experimental parts is clearly written as the entire paper. The paper is ready for publication in Nanomaterials.

Round 2

Reviewer 1 Report

The Reviewer would like to thank the authors for the clarifications that have been addressed in the revised version of the manuscript, and recommends the present study for publication.